# Deep Voice 2: Multi-Speaker Neural Text-to-Speech

**Sercan Ö. Arık***
sercanarik@baidu.com

**Gregory Diamos***
gregdiamos@baidu.com

**Andrew Gibiansky***
gibianskyandrew@baidu.com

**John Miller***
millerjohn@baidu.com

**Kainan Peng***
pengkainan@baidu.com

**Wei Ping***
pingwei01@baidu.com

**Jonathan Raiman***
jonathanraiman@baidu.com

**Yanqi Zhou***
zhouyanqi@baidu.com

**Baidu Silicon Valley Artificial Intelligence Lab**
1195 Bordeaux Dr. Sunnyvale, CA 94089

## Abstract

We introduce a technique for augmenting neural text-to-speech (TTS) with low-dimensional trainable speaker embeddings to generate different voices from a single model. As a starting point, we show improvements over the two state-of-the-art approaches for single-speaker neural TTS: Deep Voice 1 and Tacotron. We introduce Deep Voice 2, which is based on a similar pipeline with Deep Voice 1, but constructed with higher performance building blocks and demonstrates a significant audio quality improvement over Deep Voice 1. We improve Tacotron by introducing a post-processing neural vocoder, and demonstrate a significant audio quality improvement. We then demonstrate our technique for multi-speaker speech synthesis for both Deep Voice 2 and Tacotron on two multi-speaker TTS datasets. We show that a single neural TTS system can learn hundreds of unique voices from less than half an hour of data per speaker, while achieving high audio quality synthesis and preserving the speaker identities almost perfectly.

## 1 Introduction

Artificial speech synthesis, commonly known as text-to-speech (TTS), has a variety of applications in technology interfaces, accessibility, media, and entertainment. Most TTS systems are built with a single speaker voice, and multiple speaker voices are provided by having distinct speech databases or model parameters. As a result, developing a TTS system with support for multiple voices requires much more data and development effort than a system which only supports a single voice.

In this work, we demonstrate that we can build all-neural multi-speaker TTS systems which share the vast majority of parameters between different speakers. We show that not only can a single model generate speech from multiple different voices, but also that significantly less data is required per speaker than when training single-speaker systems.

Concretely, we make the following contributions:

1. We present Deep Voice 2, an improved architecture based on Deep Voice 1 (Arik et al., 2017).
2. We introduce a WaveNet-based (Oord et al., 2016) spectrogram-to-audio neural vocoder, and use it with Tacotron (Wang et al., 2017) as a replacement for Griffin-Lim audio generation.

3. Using these two single-speaker models as a baseline, we demonstrate multi-speaker neural speech synthesis by introducing trainable speaker embeddings into Deep Voice 2 and Tacotron.

We organize the rest of this paper as follows. Section 2 discusses related work and what makes the contributions of this paper distinct from prior work. Section 3 presents Deep Voice 2 and highlights the differences from Deep Voice 1. Section 4 explains our speaker embedding technique for neural TTS models and shows multi-speaker variants of the Deep Voice 2 and Tacotron architectures. Section 5.1 quantifies the improvement for single speaker TTS through a mean opinion score (MOS) evaluation and Section 5.2 presents the synthesized audio quality of multi-speaker Deep Voice 2 and Tacotron via both MOS evaluation and a multi-speaker discriminator accuracy metric. Section 6 concludes with a discussion of the results and potential future work.

## 2   Related Work

We discuss the related work relevant to each of our claims in Section 1 in order, starting from single-speaker neural speech synthesis and moving on to multi-speaker speech synthesis and metrics for generative model quality.

With regards to single-speaker speech synthesis, deep learning has been used for a variety of subcomponents, including duration prediction (Zen et al., 2016), fundamental frequency prediction (Ronanki et al., 2016), acoustic modeling (Zen and Sak, 2015), and more recently autoregressive sample-by-sample audio waveform generation (e.g., Oord et al., 2016; Mehri et al., 2016). Our contributions build upon recent work in entirely neural TTS systems, including Deep Voice 1 (Arik et al., 2017), Tacotron (Wang et al., 2017), and Char2Wav (Sotelo et al., 2017). While these works focus on building single-speaker TTS systems, our paper focuses on extending neural TTS systems to handle multiple speakers with less data per speaker.

Our work is not the first to attempt a multi-speaker TTS system. For instance, in traditional HMM-based TTS synthesis (e.g., Yamagishi et al., 2009), an average voice model is trained using multiple speakers' data, which is then adapted to different speakers. DNN-based systems (e.g., Yang et al., 2016) have also been used to build average voice models, with i-vectors representing speakers as additional inputs and separate output layers for each target speaker. Similarly, Fan et al. (2015) uses a shared hidden representation among different speakers with speaker-dependent output layers predicting vocoder parameters (e.g., line spectral pairs, aperiodicity parameters etc.). For further context, Wu et al. (2015) empirically studies DNN-based multi-speaker modeling. More recently, speaker adaptation has been tackled with generative adversarial networks (GANs) (Hsu et al., 2017).

We instead use trainable speaker embeddings for multi-speaker TTS. The approach was investigated in speech recognition (Abdel-Hamid and Jiang, 2013), but is a novel technique in speech synthesis. Unlike prior work which depends on fixed embeddings (e.g. i-vectors), the speaker embeddings used in this work are trained jointly with the rest of the model from scratch, and thus can directly learn the features relevant to the speech synthesis task. In addition, this work does not rely on per-speaker output layers or average voice modeling, which leads to higher-quality synthesized samples and lower data requirements (as there are fewer unique parameters per speaker to learn).

In order to evaluate the distinctiveness of the generated voices in an automated way, we propose using the classification accuracy of a speaker discriminator. Similar metrics such as an "Inception score" have been used for quantitative quality evaluations of GANs for image synthesis (e.g., Salimans et al., 2016). Speaker classification has been studied with both traditional GMM-based methods (e.g., Reynolds et al., 2000) and more recently with deep learning approaches (e.g., Li et al., 2017).

## 3   Single-Speaker Deep Voice 2

In this section, we present Deep Voice 2, a neural TTS system based on Deep Voice 1 (Arik et al., 2017). We keep the general structure of the Deep Voice 1 (Arik et al., 2017), as depicted in Fig. 1 (the corresponding training pipeline is depicted in Appendix A). Our primary motivation for presenting an improved single-speaker model is to use it as the starting point for a high-quality multi-speaker model.

One major difference between Deep Voice 2 and Deep Voice 1 is the separation of the phoneme duration and frequency models. Deep Voice 1 has a single model to jointly predict phoneme duration

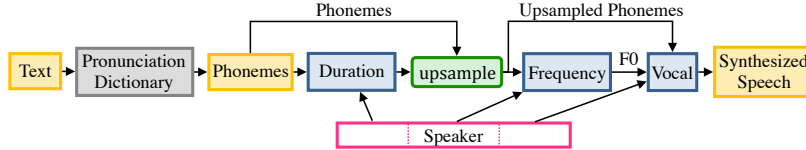

Figure 1: Inference system diagram: first text-phonemes dictionary conversion, second predict phoneme durations, third upsample and generate $F_0$, finally feed $F_0$ and phonemes to vocal model.

and frequency profile (voicedness and time-dependent fundamental frequency, $F_0$). In Deep Voice 2, the phoneme durations are predicted first and then are used as inputs to the frequency model.

In the subsequent subsections, we present the models used in Deep Voice 2. All models are trained separately using the hyperparameters specified in Appendix B. We will provide a quantitative comparison of Deep Voice 1 and Deep Voice 2 in Section 5.1.

### 3.1 Segmentation model

Estimation of phoneme locations is treated as an unsupervised learning problem in Deep Voice 2, similar to Deep Voice 1. The segmentation model is convolutional-recurrent architecture with connectionist temporal classification (CTC) loss (Graves et al., 2006) applied to classify phoneme pairs, which are then used to extract the boundaries between them. The major architecture changes in Deep Voice 2 are the addition of batch normalization and residual connections in the convolutional layers. Specifically, Deep Voice 1's segmentation model computes the output of each layer as

$$h^{(l)} = \text{relu}\left(W^{(l)} * h^{(l-1)} + b^{(l)}\right), \tag{1}$$

where $h^{(l)}$ is the output of the $l$-th layer, $W^{(l)}$ is the convolution filterbank, $b^{(l)}$ is the bias vector, and $*$ is the convolution operator. In contrast, Deep Voice 2's segmentation model layers instead compute

$$h^{(l)} = \text{relu}\left(h^{(l-1)} + \text{BN}\left(W^{(l)} * h^{(l-1)}\right)\right), \tag{2}$$

where BN is batch normalization (Ioffe and Szegedy, 2015). In addition, we find that the segmentation model often makes mistakes for boundaries between silence phonemes and other phonemes, which can significantly reduce segmentation accuracy on some datasets. We introduce a small post-processing step to correct these mistakes: whenever the segmentation model decodes a silence boundary, we adjust the location of the boundary with a silence detection heuristic.[2]

### 3.2 Duration Model

In Deep Voice 2, instead of predicting a continuous-valued duration, we formulate duration prediction as a sequence labeling problem. We discretize the phoneme duration into log-scaled buckets, and assign each input phoneme to the bucket label corresponding to its duration. We model the sequence by a conditional random field (CRF) with pairwise potentials at output layer (Lample et al., 2016). During inference, we decode discretized durations from the CRF using the Viterbi forward-backward algorithm. We find that quantizing the duration prediction and introducing the pairwise dependence implied by the CRF improves synthesis quality.

### 3.3 Frequency Model

After decoding from the duration model, the predicted phoneme durations are upsampled from a per-phoneme input features to a per-frame input for the frequency model.[3] Deep Voice 2 frequency

model consists of multiple layers: firstly, bidirectional gated recurrent unit (GRU) layers (Cho et al., 2014) generate hidden states from the input features. From these hidden states, an affine projection followed by a sigmoid nonlinearity produces the probability that each frame is voiced. Hidden states are also used to make two separate normalized $F_0$ predictions. The first prediction, $f_{\text{GRU}}$, is made with a single-layer bidirectional GRU followed by an affine projection. The second prediction, $f_{\text{conv}}$, is made by adding up the contributions of multiple convolutions with varying convolution widths and a single output channel. Finally, the hidden state is used with an affine projection and a sigmoid nonlinearity to predict a mixture ratio $\omega$, which is used to weigh the two normalized frequency predictions and combine them into

$$f = \omega \cdot f_{\text{GRU}} + (1 - \omega) \cdot f_{\text{conv}}. \tag{3}$$

The normalized prediction $f$ is then converted to the true frequency $F_0$ prediction via

$$F_0 = \mu_{F_0} + \sigma_{F_0} \cdot f, \tag{4}$$

where $\mu_{F_0}$ and $\sigma_{F_0}$ are, respectively, the mean and standard deviation of $F_0$ for the speaker the model is trained on. We find that predicting $F_0$ with a mixture of convolutions and a recurrent layer performs better than predicting with either one individually. We attribute this to the hypothesis that including the wide convolutions reduces the burden for the recurrent layers to maintain state over a large number of input frames, while processing the entire context information efficiently.

### 3.4 Vocal Model

The Deep Voice 2 vocal model is based on a WaveNet architecture (Oord et al., 2016) with a two-layer bidirectional QRNN (Bradbury et al., 2017) conditioning network, similar to Deep Voice 1. However, we remove the $1 \times 1$ convolution between the gated tanh nonlinearity and the residual connection. In addition, we use the same conditioner bias for every layer of the WaveNet, instead of generating a separate bias for every layer as was done in Deep Voice 1. [4]

## 4  Multi-Speaker Models with Trainable Speaker Embeddings

In order to synthesize speech from multiple speakers, we augment each of our models with a single low-dimensional speaker embedding vector per speaker. Unlike previous work, our approach does not rely on per-speaker weight *matrices* or *layers*. Speaker-dependent parameters are stored in a very low-dimensional vector and thus there is near-complete weight sharing between speakers. We use speaker embeddings to produce recurrent neural network (RNN) initial states, nonlinearity biases, and multiplicative gating factors, used throughout the networks. Speaker embeddings are initialized randomly with a uniform distribution over $[-0.1, 0.1]$ and trained jointly via backpropagation; each model has its own set of speaker embeddings.

To encourage each speaker's unique voice signature to influence the model, we incorporate the speaker embeddings into multiple portions of the model. Empirically, we find that simply providing the speaker embeddings to the input layers does not work as well for any of the presented models besides the vocal model, possibly due to the high degree of residual connections present in the WaveNet and due to the difficulty of learning high-quality speaker embeddings. We observed that several patterns tend to yield high performance:

- **Site-Specific Speaker Embeddings:** For every use site in the model architecture, transform the shared speaker embedding to the appropriate dimension and form through an affine projection and a nonlinearity.

- **Recurrent Initialization:** Initialize recurrent layer hidden states with site-specific speaker embeddings.

- **Input Augmentation:** Concatenate a site-specific speaker embedding to the input at every timestep of a recurrent layer.

- **Feature Gating:** Multiply layer activations elementwise with a site-specific speaker embedding to render adaptable information flow. [5]

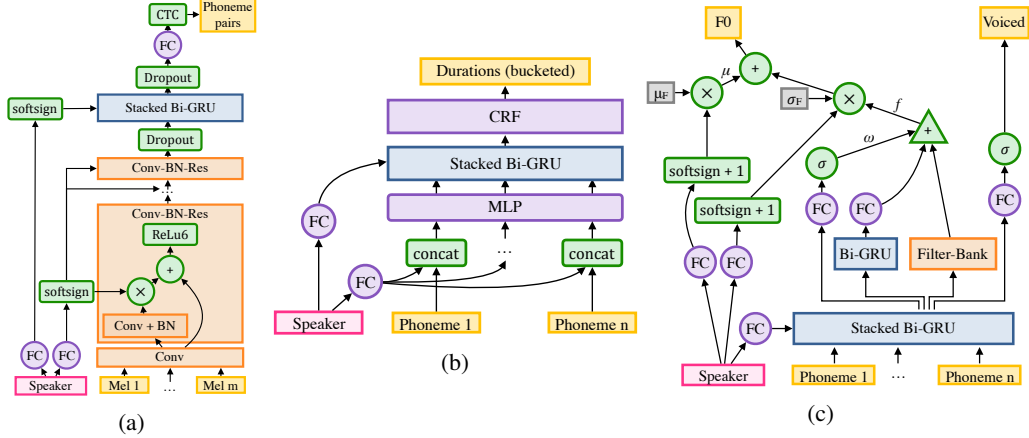

Figure 2: Architecture for the multi-speaker (a) segmentation, (b) duration, and (c) frequency model.

Next, we describe how speaker embeddings are used in each architecture.

## 4.1 Multi-Speaker Deep Voice 2

The Deep Voice 2 models have separate speaker embeddings for each model. Yet, they can be viewed as chunks of a larger speaker embedding, which are trained independently.

### 4.1.1 Segmentation Model

In multi-speaker segmentation model, we use feature gating in the residual connections of the convolution layers. Instead of Eq. (2), we multiply the batch-normalized activations by a site-specific speaker embedding:

$$h^{(l)} = \text{relu}\left(h^{(l-1)} + \text{BN}\left(W * h^{(l-1)}\right) \cdot g_s\right), \tag{5}$$

where $g_s$ is a site-specific speaker embedding. The same site-specific embedding is shared for all the convolutional layers. In addition, we initialize each of the recurrent layers with a second site specific embedding. Similarly, each layer shares the same site-specific embedding, rather than having a separate embedding per layer.

### 4.1.2 Duration Model

The multi-speaker duration model uses speaker-dependent recurrent initialization and input augmentation. A site-specific embedding is used to initialize RNN hidden states, and another site-specific embedding is provided as input to the first RNN layer by concatenating it to the feature vectors.

### 4.1.3 Frequency Model

The multi-speaker frequency model uses recurrent initialization, which initializes the recurrent layers (except for the recurrent output layer) with a single site-specific speaker-embedding. As described in Section 3.3, the recurrent and convolutional output layers in the single-speaker frequency model predict a *normalized* frequency, which is then converted into the true $F_0$ by a fixed linear transformation. The linear transformation depends on the mean and standard deviation of observed $F_0$ for the speaker. These values vary greatly between speakers: male speakers, for instance, tend to have a much lower mean $F_0$. To better adapt to these variations, we make the mean and standard deviation trainable model parameters and multiply them by scaling terms which depend on the speaker embeddings. Specifically, instead of Eq. (4), we compute the $F_0$ prediction as

$$F_0 = \mu_{F_0} \cdot \left(1 + \text{softsign}\left(V_\mu{}^T g_f\right)\right) + \sigma_{F_0} \cdot \left(1 + \text{softsign}\left(V_\sigma{}^T g_f\right)\right) \cdot f, \tag{6}$$

where $g_f$ is a site-specific speaker embedding, $\mu_{F_0}$ and $\sigma_{F_0}$ are trainable scalar parameters initialized to the $F_0$ mean and standard deviation on the dataset, and $V_\mu$ and $V_\sigma$ are trainable parameter vectors.

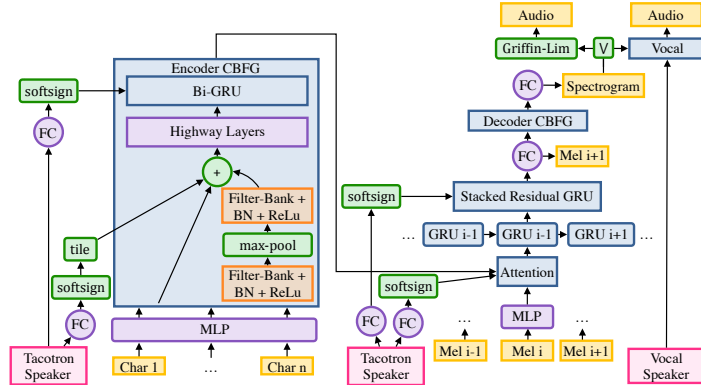

Figure 3: Tacotron with speaker conditioning in the Encoder CBHG module and decoder with two ways to convert spectrogram to audio: Griffin-Lim or our speaker-conditioned Vocal model.

### 4.1.4 Vocal Model

The multi-speaker vocal model uses only input augmentation, with the site-specific speaker embedding concatenated onto each input frame of the conditioner. This differs from the global conditioning suggested in Oord et al. (2016) and allows the speaker embedding to influence the local conditioning network as well.

Without speaker embeddings, the vocal model is still able to generate somewhat distinct-sounding voices because of the disctinctive features provided by the frequency and duration models. Yet, having speaker embeddings in the vocal model increases the audio quality. We indeed observe that the embeddings converge to a meaningful latent space.

## 4.2 Multi-Speaker Tacotron

In addition to extending Deep Voice 2 with speaker embeddings, we also extend Tacotron (Wang et al., 2017), a sequence-to-sequence character-to-waveform model. When training multi-speaker Tacotron variants, we find that model performance is highly dependent on model hyperparameters, and that some models often fail to learn attention mechanisms for a small subset of speakers. We also find that if the speech in each audio clip does not start at the same timestep, the models are much less likely to converge to a meaningful attention curve and recognizable speech; thus, we trim all initial and final silence in each audio clip. Due to the sensitivity of the model to hyperparameters and data preprocessing, we believe that additional tuning may be necessary to obtain maximal quality. Thus, our work focuses on demonstrating that Tacotron, like Deep Voice 2, is capable of handling multiple speakers through speaker embeddings, rather than comparing the quality of the two architectures.

### 4.2.1 Character-to-Spectrogram Model

The Tacotron character-to-spectrogram architecture consists of a convolution-bank-highway-GRU (CBHG) encoder, an attentional decoder, and a CBHG post-processing network. Due to the complexity of the architecture, we leave out a complete description and instead focus on our modifications.

We find that incorporating speaker embeddings into the CBHG post-processing network degrades output quality, whereas incorporating speaker embeddings into the character encoder is necessary. Without a speaker-dependent CBHG encoder, the model is incapable of learning its attention mechanism and cannot generate meaningful output (see Appendix D.2 for speaker-dependent attention visualizations). In order to condition the encoder on the speaker, we use one site-specific embedding as an extra input to each highway layer at each timestep and initialize the CBHG RNN state with a second site-specific embedding.

We also find that augmenting the decoder with speaker embeddings is helpful. We use one site-specific embedding as an extra input to the decoder pre-net, one extra site-specific embedding as the initial attention context vector for the attentional RNN, one site-specific embedding as the initial decoder GRU hidden state, and one site-specific embedding as a bias to the tanh in the content-based attention mechanism.

| Model | Samp. Freq. | MOS |
|---|---|---|
| Deep Voice 1 | 16 KHz | $2.05 \pm 0.24$ |
| Deep Voice 2 | 16 KHz | $2.96 \pm 0.38$ |
| Tacotron (Griffin-Lim) | 24 KHz | $2.57 \pm 0.28$ |
| Tacotron (WaveNet) | 24 KHz | $4.17 \pm 0.18$ |

Table 1: Mean Opinion Score (MOS) evaluations with 95% confidence intervals of Deep Voice 1, Deep Voice 2, and Tacotron. Using the crowdMOS toolkit, batches of samples from these models were presented to raters on Mechanical Turk. Since batches contained samples from all models, the experiment naturally induces a comparison between the models.

### 4.2.2 Spectrogram-to-Waveform Model

The original Tacotron implementation in (Wang et al., 2017) uses the Griffin-Lim algorithm to convert spectrograms to time-domain audio waveforms by iteratively estimating the unknown phases.[6] We observe that minor noise in the input spectrogram causes noticeable estimation errors in the Griffin-Lim algorithm and the generated audio quality is degraded. To produce higher quality audio using Tacotron, instead of using Griffin-Lim, we train a WaveNet-based neural vocoder to convert from linear spectrograms to audio waveforms. The model used is equivalent to the Deep Voice 2 vocal model, but takes linear-scaled log-magnitude spectrograms instead of phoneme identity and $F_0$ as input. The combined Tacotron-WaveNet model is shown in Fig. 3. As we will show in Section 5.1, WaveNet-based neural vocoder indeed significantly improves single-speaker Tacotron as well.

## 5 Results

In this section, we will present the results on both single-speaker and multi-speaker speech synthesis using the described architectures. All model hyperparameters are presented in Appendix B.

### 5.1 Single-Speaker Speech Synthesis

We train Deep Voice 1, Deep Voice 2, and Tacotron on an internal English speech database containing approximately 20 hours of single-speaker data. The intermediate evaluations of models in Deep Voice 1 and Deep Voice 2 can be found in Table 3 within Appendix A. We run an MOS evaluation using the crowdMOS framework (Ribeiro et al., 2011) to compare the quality of samples (Table 1). The results show conclusively that the architecture improvements in Deep Voice 2 yield significant gains in quality over Deep Voice 1. They also demonstrate that converting Tacotron-generated spectrograms to audio using WaveNet is preferable to using the iterative Griffin-Lim algorithm.

### 5.2 Multi-Speaker Speech Synthesis

We train all the aforementioned models on the VCTK dataset with 44 hours of speech, which contains 108 speakers with approximately 400 utterances each. We also train all models on an internal dataset of audiobooks, which contains 477 speakers with 30 minutes of audio each (for a total of $\sim$238 hours). The consistent sample quality observed from our models indicates that our architectures can easily learn hundreds of distinct voices with a variety of different accents and cadences. We also observe that the learned embeddings lie in a meaningful latent space (see Fig. 4 as an example and Appendix D for more details).

In order to evaluate the quality of the synthesized audio, we run MOS evaluations using the crowdMOS framework, and present the results in Table 2. We purposefully include ground truth samples in the set being evaluated, because the accents in datasets are likely to be unfamiliar to our North American crowdsourced raters and will thus be rated poorly due to the accent rather than due to the model quality. By including ground truth samples, we are able to compare the MOS of the models with the ground truth MOS and thus evaluate the model quality rather than the data quality; however, the resulting MOS may be lower, due to the implicit comparison with the ground truth samples. Overall, we observe that the Deep Voice 2 model can approach an MOS value that is close to the ground truth, when low sampling rate and companding/expanding taken into account.

| Dataset | Multi-Speaker Model | Samp. Freq. | MOS | Acc. |
|---|---|---|---|---|
| VCTK | Deep Voice 2 (20-layer WaveNet) | 16 KHz | 2.87±0.13 | 99.9% |
| VCTK | Deep Voice 2 (40-layer WaveNet) | 16 KHz | 3.21±0.13 | 100 % |
| VCTK | Deep Voice 2 (60-layer WaveNet) | 16 KHz | 3.42±0.12 | 99.7% |
| VCTK | Deep Voice 2 (80-layer WaveNet) | 16 KHz | 3.53±0.12 | 99.9% |
| VCTK | Tacotron (Griffin-Lim) | 24 KHz | 1.68±0.12 | 99.4% |
| VCTK | Tacotron (20-layer WaveNet) | 24 KHz | 2.51±0.13 | 60.9% |
| VCTK | Ground Truth Data | 48 KHz | 4.65±0.06 | 99.7% |
| Audiobooks | Deep Voice 2 (80-layer WaveNet) | 16 KHz | 2.97±0.17 | 97.4% |
| Audiobooks | Tacotron (Griffin-Lim) | 24 KHz | 1.73±0.22 | 93.9% |
| Audiobooks | Tacotron (20-layer WaveNet) | 24 KHz | 2.11±0.20 | 66.5% |
| Audiobooks | Ground Truth Data | 44.1 KHz | 4.63±0.04 | 98.8% |

Table 2: MOS and classification accuracy for all multi-speaker models. To obtain MOS, we use crowdMOS toolkit as detailed in Table 1. We also present classification accuracies of the speaker discriminative models (see Appendix E for details) on the samples, showing that the synthesized voices are as distinguishable as ground truth audio.

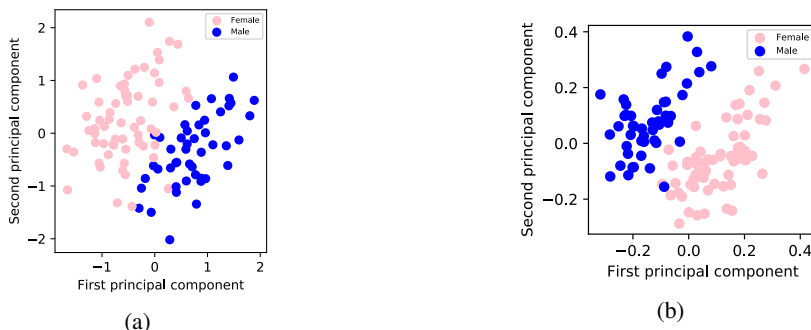

(a)                                     (b)

Figure 4: Principal components of the learned speaker embeddings for the (a) 80-layer vocal model and (b) character-to-spectrogram model for VCTK dataset. See Appendix D.3 for details.

A multi-speaker TTS system with high sample quality but indistinguishable voices would result in high MOS, but fail to meet the desired objective of reproducing the input voices accurately. To show that our models not only generate high quality samples, but also generate *distinguishable* voices, we also measure the classification accuracy of a speaker discriminative model on our generated samples. The speaker discriminative is a convolutional network trained to classify utterances to their speakers, trained on the same dataset as the TTS systems themselves. If the voices were indistinguishable (or the audio quality was low), the classification accuracy would be much lower for synthesized samples than it is for the ground truth samples. As we demonstrate in Table 2, classification accuracy demonstrates that samples generated from our models are as distinguishable as the ground truth samples (see Appendix E for more details). The classification accuracy is only significantly lower for Tacotron with WaveNet, and we suspect that generation errors in the spectrogram are exacerbated by the WaveNet, as it is trained with ground truth spectrograms.

# 6    Conclusion

In this work, we explore how entirely-neural speech synthesis pipelines may be extended to multi-speaker text-to-speech via low-dimensional trainable speaker embeddings. We start by presenting Deep Voice 2, an improved single-speaker model. Next, we demonstrate the applicability of our technique by training both multi-speaker Deep Voice 2 and multi-speaker Tacotron models, and evaluate their quality through MOS. In conclusion, we use our speaker embedding technique to create high quality text-to-speech systems and conclusively show that neural speech synthesis models can learn effectively from small amounts of data spread among hundreds of different speakers.

The results presented in this work suggest many directions for future research. Future work may test the limits of this technique and explore how many speakers these models can generalize to, how little data is truly required per speaker for high quality synthesis, whether new speakers can be added to a system by fixing model parameters and solely training new speaker embeddings, and whether the speaker embeddings can be used as a meaningful vector space, as is possible with word embeddings.

## Footnotes

*Listed alphabetically.

[2]We compute the smoothed normalized audio power as $p[n] = (x[n]^2/x_{\max}^2) * g[n]$, where $x[n]$ is the audio signal, $g[n]$ is the impulse response of a Gaussian filter, $x_{\max}$ is the maximum value of $x[n]$ and $*$ is one-dimensional convolution operation. We assign the silence phoneme boundaries when $p[n]$ exceeds a fixed threshold. The optimal parameter values for the Gaussian filter and the threshold depend on the dataset and audio sampling rate.

[3]Each frame is ensured to be 10 milliseconds. For example, if a phoneme lasts 20 milliseconds, the input features corresponding to that phoneme will be repeated in 2 frames. If it lasts less than 10 milliseconds, it is extend to a single frame.

[4]We find that these changes reduce model size by a factor of $\sim 7$ and speed up inference by $\sim 25\%$, while yielding no perceptual change in quality. However, we do not focus on demonstrating these claims in this paper.

[5]We hypothesize that feature gating lets the model learn the union of all necessary features while allowing speaker embeddings to determine what features are used for each speaker and how much influence they will have on the activations.

[6]Estimation of the unknown phases is done by repeatedly converting between frequency and time domain representations of the signal using the short-time Fourier transform and its inverse, substituting the magnitude of each frequency component to the predicted magnitude at each step.

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
