[Reviews · NeurIPS 2017]

Reviewer 1



Seems like this paper will attract a lot of interest. Not doing a detailed review since it looks like it will be a definite accept.

Reviewer 2



This paper presents Deep Voice 2, and neural TTS system that is able to generate voices from multiple speakers. This is a solid paper. The paper would benefit from addressing a few points: - there is no mention of training and, in particular, inference time. The original WaveNet is notably slow, how do these systems perform? - why use different sampling frequencies for different systems? I don't believe there is a technical hurdle in using the same sampling frequency for all, and it would be easier to compare MOS scores. - the paper mentions that giving the ground truth may influence the MOS for other methods: why not test that by splitting raters into two groups, those that get to listen to the ground truth and those that don't, and comparing the results?

Reviewer 3



This paper presents a solid piece of work on the speaker-dependent neural TTS system, building on previous works of Deep Voice and Tacotron architecture. The key idea is to learn a speaker-dependent embedding vector jointly with the neural TTS model. The paper is clearly written, and the experiments are presented well. My comments are as follows. -- the speaker embedding vector approach is very similar to the speaker code approach for speaker adaptation studied in ASR, e.g., Ossama Abdel-Hamid, Hui Jiang, "Fast speaker adaptation of hybrid NN/HMM model for speech recognition based on discriminative learning of speaker code", ICASSP 2013 This should be mentioned in the paper. ASR researchers later find that using fixed speaker embeddings such i-vectors can work equally well (or even better). It would be interesting if the authors could present some experimental results on that. The benefit is that you may be able to do speaker adaptation fairly easily, without learning the speaker embedding as the presented approach. --in the second of paragraph of Sec. 3, you mentioned that the phoneme duration and frequency models are separated in this work. Can you explain your motivation? --the CTC paper from A. Graves should be cited in section 3.1. Also, CTC is not a good alignment model for ASR, as it is trained by marginalizing over all the possible alignments. For ASR, the alignments from CTC may be very biased from the ground truth. I am curious how much segmentation error do you have, and have you tried some other segmentation models, e.g., segmental RNN? --in Table 1, can you present comparisons between Deep Voice and Tocotron with the same sample frequency? --is Deep Voice 2 trained end-to-end, or the modules are trained separately? It would be helpful to readers to clarify that. --the speaker embedding are used as input to several different modules, some ablation experiments in the appendix would be helpful to see how much difference it would make for each module.